# Use of Tobacco and Nicotine Products among Young People in Denmark—Status in Single and Dual Use

**DOI:** 10.3390/ijerph19095623

**Published:** 2022-05-05

**Authors:** Lotus Sofie Bast, Marie Borring Klitgaard, Simone Gad Kjeld, Nanna Schneekloth Jarlstrup, Anne Illemann Christensen

**Affiliations:** National Institute of Public Health, University of Southern Denmark, Studiestraede 6, 1455 Copenhagen, Denmark; mboa@sdu.dk (M.B.K.); simk@sdu.dk (S.G.K.); nasc@sdu.dk (N.S.J.); anch@sdu.dk (A.I.C.)

**Keywords:** tobacco, cigarette, nicotine product, e-cigarette, smokeless tobacco, snus, nicotine pouches, heated tobacco, dual use, youth, adolescents, young adults, tobacco prevention

## Abstract

Lots of new tobacco or nicotine products are being launched, e.g., e-cigarettes and smokeless tobacco, which appeal especially to the youngest part of the population. For example, the use of smokeless tobacco among Danish youth rose from approx. 2% in 2010 to 9% in 2020. Hence, there is an urgent need to follow and intervene against youth tobacco or nicotine product use. This study explored the current use of cigarettes, e-cigarettes, heated tobacco, and smokeless tobacco among Danish 15- to 29-year-olds. Further, we examined the concurrent use of two products or more. We used a nationwide survey conducted among 15- to 29-year-olds in February and March 2020. Overall, approx. 35,700 individuals received the questionnaire of which 35.5% responded (*n* = 13,315). One out of five (20.1%) smoked cigarettes, half of them daily, the other half occasionally. About one in twenty (3.9%) used e-cigarettes (daily or occasionally), and more than one in three (31.6%) had tried e-cigarettes. The use of heated tobacco among Danish youth is still relatively limited (0.3%). In comparison, about 9% used smokeless tobacco (daily or occasionally). Overall, 27.0% stated that they use at least one type of tobacco or nicotine product, while 5.6% used more than one product. Monitoring tobacco-related behavior in youth provides extremely important information for, e.g., policymakers and health professionals.

## 1. Introduction

The global tobacco market has expanded widely in the last decade; a range of new tobacco- or nicotine-containing products have been introduced as alternatives to cigarettes—with increasing popularity, especially among youth in the Western world [1,2,3]. Moreover, international trends show a rise in the use of various tobacco or nicotine products simultaneously, i.e., dual and multiple use [4,5]. In Denmark, these current trends among youth have received significant attention from politicians and have raised public health concerns; Decades of decreasing smoking prevalence among the general population was in 2017 replaced by stagnation, and in certain groups of the Danish youth, even an increase; e.g., among 16- to 25-year-olds [6,7]. However, a recent Danish report showed that daily—but not occasional—smoking among 18–19-year-olds decreased again [8]—whether this decline is connected to a concurrent increase in the use of other tobacco or nicotine products has not yet been addressed.

Internationally, in some high-income countries, the use of other tobacco or nicotine-containing products has exceeded the use of cigarettes among the youth population [3,9,10]. Further, the use of more than one tobacco or nicotine product is becoming prevalent. For example, among U.S. high school students in 2021, the most commonly cited product was e-cigarettes, which are currently used by 11.3% of high school students. Followed by cigarettes, cigars, smokeless tobacco, hookahs, nicotine pouches, heated tobacco products, and pipe tobacco. Overall, 13.4% were current (past 30 days) users of any product [10]. In a sample of adolescents (13- to 17-years) in Norway—a country similar to Denmark living wise, both culturally and economically—31.0% said that they had tried at least one tobacco product; among them, snus was twice as prevalent as both cigarettes and e-cigarettes [3].

Though the shift from cigarettes to other tobacco products may cause a decrease in cigarette smoking, the use of any tobacco product constitutes a threat to both the individual and the overall public health; the hazards of not only tobacco but also nicotine have become more evident in recent years, i.e., it causes increased risk in cardiovascular and respiratory illnesses, and impact the reproductive system [11]. Moreover, nicotine negatively impacts the development of the adolescent brain [12,13]. Further, experimenting with products such as e-cigarettes in youth may work as a gateway to cigarette smoking [14,15]. E-cigarettes, smokeless tobacco (i.e., snus, chewing tobacco, and nicotine pouches), and heated tobacco products are among the products that seem to gain increasing popularity among the youth worldwide [16]. A recent Danish report sought to map out the use of these products among the general population in Denmark [17]—however, the use among Danish youth is still not well-documented. Neither is the concurrent use of two or more of the products.

Although, for years, tobacco regulations have been lenient in Denmark, a new law went into effect. By 1 April 2020, tobacco prices in Denmark increased from approx. DKK 40 to 55 for a package with 20 cigarettes (corresponding to a change from USD 6 to 8). In 2022, the prices will be increased by an additional DKK 5 (approx. USD 1) so that the overall increase in prices from 2020 to 2022 will be 50%. Furthermore, by 1 January 2021, a new law with multiple tobacco preventive initiatives became effective. The law comprises a range of initiatives; among them are a ban on additives flavors such as fruit, menthol, and mint flavor; a ban on promotion at the point of sale (POS); standardized tobacco packaging; health warnings on all nicotine-containing products; increased age control at POS; smoke-free school time; and an extended ban against advertising and sponsorship of tobacco, e-cigarettes, and nicotine-containing products.

In response to the current concerns in terms of youth tobacco and nicotine product use, as well as the limited in-depth knowledge of concurrent product use, a nationwide Danish study entitled “§SMOKE—A study of tobacco, behavior and regulations” was initiated. The baseline survey was conducted before the implementation of any of the new regulations and hence will be used as a comparison when following trends in the years to come. The aims of the §SMOKE study are to follow the tobacco and nicotine product use among the Danish youth as well evaluate the effect of the stronger regulations with repeated cross-sectional surveys among 15-to-29-year-olds in the years 2020 until 2025.

This study reports on current tobacco or nicotine product use (i.e., use of cigarettes, e-cigarettes, smokeless tobacco, and heated tobacco) and concurrent use of products in 2020.

## 2. Materials and Methods

### 2.1. Study Design

This study is based on baseline data from “§SMOKE—A study of tobacco, behavior and regulations”. The baseline survey is part of a larger evaluation running from 2020 until 2025. The National Institute of Public Health is responsible for the data collection with funding from the Danish Health Authority and the TrygFoundation. The repeated yearly data collections in the years from 2021 until 2025 are carried out by the National Institute of Public Health in collaboration with The Danish Cancer Society, The Heart Association, and The Lung Association, with financial support from the TrygFoundation.

### 2.2. Participants and Sample Size

Study participants were a nationally representative sample of Danish 15–29-year-olds with permanent residence in Denmark at the time of study. All Danish citizens have a unique identification number registered in The Danish Civil Registration System that was used to draw the random sample [18].

The current smoking prevalence among Danish youth at the time of study was 26% (daily or occasionally) [17]. To ensure that essential analyses could be made, including investigating changes in tobacco patterns over the study period as well as subgroup- and stratified analyses, the requested sample size was set to 15,000 respondents. With an expected response rate in this age group at 40%, we invited 37,500 persons to the study [19].

### 2.3. Data Collection

Data collection was conducted from February to March 2020. The majority of participants (98%) received a secure electronic e-mail with a link to the survey, while two percent of participants received a postal letter with a paper questionnaire, including a link to the web survey. Two reminders were sent to all non-responders. Topics in the questionnaire were related to the patterns of tobacco and nicotine product use as well as topics specifically related to three main components of the new tobacco law to reduce tobacco uptake among the youth in Denmark, especially with focus towards increased tobacco prices, POS display ban, and standardized packaging [14]. In total, 37,482 individuals received the questionnaire of which 13,315 returned valid responses (response rate = 35.5%), see Figure 1: Flow diagram.

### 2.4. Sociodemographic Characteristics and Weighting Procedure

Weights were constructed using auxiliary information from Statistics Denmark’s registers to account for the possible selection bias in which participants responded to the survey. These weights were based on information on gender and age, which were the primary factors that differed between respondents and non-respondents (See also Table 1). Further, we examined the distribution of responses and non-responses according to region in Denmark and found that the distributions did not markedly differ. The weights ensured representability of the study sample and reduced the impact of non-responses (for more detailed information, see also [19]).

### 2.5. Measures

We received data on gender and age from the Civil Registration System [18]. The remaining variables used for this study were obtained by participants’ self-reported answers to the questionnaire. In Table 2, an overview of the variables used for the current study is shown, including the item, response categories, and coding.

### 2.6. Analyses

We examined possible differences in tobacco use (i.e., use of cigarettes, e-cigarettes, smokeless tobacco, and heated tobacco) according to age groups and gender using χ^2^-tests. A *p*-value of <0.05 was considered statistically significant. Moreover, we conducted descriptive analyses for dual use, including the proportion of the youth that daily or regularly use two or more of the included tobacco products (i.e., cigarettes, e-cigarettes, smokeless tobacco, and heated tobacco). Finally, we examined the type of tobacco or nicotine product used first stratified by gender and age groups. We used the Stata version 16 for all data analyses. Percentages presented in tables and figures are weighted, whereas the number of respondents is not. Further, numbers of five or fewer observations are reported as n/a.

## 3. Results

One-fifth (20.1%) said that they smoked cigarettes, half of them daily (Table 3). Prevalences of cigarette smoking were higher among men compared to women and most prevalent among 18–24-year-olds. In the youngest age group (15–17 years), the majority occasionally smoked, while daily smoking was most common in the oldest group (25–29 years). Overall, 3.9% stated that they used e-cigarettes (daily or occasionally), with twice as many men using e-cigarettes compared with women. Using e-cigarettes was more prevalent in the two youngest age groups, e.g., more than one-third (37.5%) of the 18- to 24-year-olds stated that they had tried e-cigarettes.

About 9% of respondents used smokeless tobacco (daily or occasionally). More men compared to women indicated to be current users, previous users, or having tried smokeless tobacco. Further, the use of smokeless tobacco was highest among the two youngest age groups (15–17 and 18–24-year-olds). The use of heated tobacco among participants was relatively limited, with less than a half percent stating to use this product. No gender differences were observed, but the use of heated tobacco was most prevalent among 18- to 24-year-olds.

Overall, more than one-fourth (27.0%) said that they used at least one type of tobacco or nicotine product (Table 3). For men, the proportion was almost one in three (31.1%), while it was around one in fourth among women (22.8%). Further, the prevalence was highest among 18- to 24-year-olds (31.3%). Moreover, a high proportion had ever tried one or more tobacco products (71.3%), with the highest prevalences detected among the oldest age group (78.1%); however, as many as 50.4% of the 15–17-year-olds reported ever use of tobacco or nicotine product.

The proportion of respondents currently using more than one product was 5.6% (data not shown). Among these, 87.2% were dual users and 12.8% were multiple product users (three or more products; Table 4). The most common combinations were cigarettes combined with e-cigarettes as well as cigarettes combined with smokeless tobacco. The combination of cigarettes and e-cigarettes was most prevalent among females, whereas the combined use of cigarettes and snus was mostly applied among men. Using more than two products was more prevalent among men compared to women (14.6 vs. 8.8%).

Most respondents said that the first product they tried was cigarettes; more than 80% in both genders (Figure 2). Overall, about 15% stated that they first tried one of the other tobacco or nicotine products. Almost one in ten men tried e-cigarettes first, and 8.4% smokeless tobacco. For women, the numbers were a little lower. No women and almost no men (too few to report) tried heated tobacco first.

Starting with e-cigarettes or smokeless tobacco was twice as prevalent among the 15- to 17-year-olds compared with the 18- to 24-year-olds (Figure 3), and almost none of the 25- to 29-year-olds started with these products (there were too few to report the specific numbers).

## 4. Discussion

The results of this study indicate that the use of tobacco and nicotine products constitutes a significant public health issue in Denmark. Overall, cigarette smoking was most prevalent among Danish youth, followed by significant use of smokeless tobacco, fewer using e-cigarettes, and very few using heated tobacco. This contrasts with findings from the US, where e-cigarette use is more prevalent than smoking among youth [20], and findings from Norway, where smokeless tobacco use has exceeded the use of cigarettes [21,22].

As shown in this study, more than one out of four (27.0%) of Danish youth are current users of at least one tobacco or nicotine-containing product, and 5.4% use two or products or more. In comparison, there were 23.6% of the 14–18-year-olds currently using at least one product in the US in 2020 [20,23]. Men are more prone than women to use more than two tobacco- or nicotine products, and the combination of dual use differed between genders; women more often used e-cigarettes and cigarettes, whereas men used smokeless tobacco and cigarettes.

Most respondents who currently use more than one product started with smoking cigarettes. However, especially among the youngest participants (15- to 17-year-olds)—a significant proportion reported first trying e-cigarettes (16.3%) and smokeless tobacco (16.2%). Among the 18- to 24-year-olds, there were also some who first initiated other products than cigarettes, but very few in the oldest age group (25- to 29-year-olds)—too few to report on actual prevalences. Obviously, the timing of introducing the different types of products influences which products are first used, i.e., e-cigarettes were just introduced to the market when respondents from the oldest age group initiated a tobacco use in their teenage years, and the smokeless product known as nicotine pouches have been on the market for only a couple of years. However, we cannot leave out the possibility of a trend towards starting with using other tobacco products than cigarettes.

If youth tobacco habits in Denmark will follow international trends, we may detect declining smoking prevalences in the coming years but also increases in the use of other tobacco products. Recent numbers among Danish adolescents indicate a decline in daily cigarette smoking among 18–19-year-olds [8]. We should pay very close attention to the development of smoking in the coming years and not at least to mechanisms such as possible gateway effects of e-cigarette use leading to cigarette smoking [24]. The obvious risk is being addicted to nicotine—however, another important mechanism is that adolescents learn the habits and rituals of smoking through using e-cigarettes, i.e., the body language, the habits of taking smoking breaks, and handling a tobacco product, which may ease the path to cigarette smoking [24]. Further, research has found that e-cigarettes themselves constitute a significant health risk [25].

In Norway, the declining smoking prevalence in youth was accompanied by the increasing use of smokeless tobacco [21,22]. Whether this trend will be adopted by the Danish youth should be followed carefully in the years to come. Smokeless tobacco, such as snus and nicotine pouches, seem to be particularly popular in Scandinavian countries. A study from 2019 found that 14% of Norwegian adolescents used snus (regularly or occasionally), and 16% had tried snus [3]. Among Finnish adolescents, 11% were current users of snus, while 9% had tried using snus [26]. Among Swedish youth (<25 years), 12.0% were current users of smokeless tobacco. In another study across 17 European countries, 1.4% on average used smokeless tobacco [27]. In our study, 9.1% were currently using smokeless tobacco, while 27.3% had tried smokeless tobacco. Thus, a significant proportion of young people in Denmark have experience with smokeless tobacco (either as current users or as having tried using it) compared to the average of youth in European countries. The current use of smokeless tobacco in Danish youth seems to correspond with current use among youth in other Scandinavian countries, e.g., Norway, Finland, and Sweden. For Norwegian youth, the use of smokeless tobacco—with or without concurrent cigarette smoking—resulted in a higher risk of adult smoking as well as using smokeless tobacco in adulthood [21].

The use of heated tobacco products in the Danish youth seems to be quite similar to other countries. In a sample of middle- and high-school students in the US, the overall percentage of ever using heated tobacco was 2.4%, and current use was 1.6% [28]. Among Korean adolescents, ever use was 2.8% [29]. In comparison, we found that ever use was 3.2% and current use 0.3%. The products are marketed as less harmful than tobacco products with the risk that especially young people find them attractive and start using them [30,31]. In a study of 16- to 19-year-olds across Canada, England, and the USA, 7.0% reported awareness of heated tobacco, and 38.6% expressed interest in trying this product [32]. Youth, who were currently smoking or had previous experience with smoking, seemed to pay more attention to heated tobacco and more interest in trying the products. Further, males, cigarette smokers, and e-cigarette users had higher susceptibility to trying heated tobacco [28,32].

### 4.1. Methodological Issues

The strengths of this study are the large sample size drawn randomly from a national register and the questionnaire with multiple tobacco and nicotine-containing products, as well as other tobacco-related items enabling the study of patterns and trends—also within subgroups. The study sample size is large enough to examine subgroups in dual tobacco product use, which most other studies are not. We examined differences among participants and non-participants and used weighting for age and gender to account for possible bias in the responses due to these factors. Further, this study is the first in a row of cross-sectional studies in the §SMOKE study, which altogether contributes to the evaluation of the tobacco regulations in Denmark.

The questionnaire was sent by secure electronic mail, which most people living in Denmark can receive. Moreover, persons without electronic mail received a postal letter. Two reminders were sent to the whole sample size to optimize the number of respondents. The response rate was 35.5%, which is in accordance with response rates among youth in other Danish population-based studies. In Denmark, we have experienced an overall trend of declining participation proportion in health surveys over the past 20–30 years, and the rather low participation might be a result of questionnaire fatigue among youth, which was also seen in other studies [33].

The reliability of self-reported survey data is based on confidence in the accuracy of the respondents’ recall as well as on their motivation to provide truthful information on the topic of interest. Youth behaviors were self-reported, with the risk of social desirability bias. However, previous research shows good correspondence between adolescent self-reported smoking status and biological measures [34,35].

Another general limitation of survey data is the cross-sectional design, which does not allow conclusions to be drawn on the direction of causality; however, the follow-ups planned in §SMOKE will allow for examinations of trends.

### 4.2. Implications

Data from the §SMOKE study provides new important knowledge, with the possibility to monitor youth tobacco and nicotine product use over a period with multiple new regulations being implemented. The questionnaires cover a wide variety of topics not included in official statistical registers.

For future research, the data derived from the §SMOKE surveys can be linked on an individual level to different official statistical registers (e.g., the Danish National Patient Register, the Danish Register of Causes of Death, The Danish National Prescription Register, and the Danish National Service Register) due to the unique personal registration numbers, which allows for analyses of the relationship between, e.g., risk factors and morbidity and mortality, social inequality in health, etc.

## 5. Conclusions

Monitoring tobacco product use in youth provides extremely important information for policymakers and health professionals. This paper shows a continued need for regulation to prevent Danish youth from initiating the use of one or more of the addictive and health-damaging tobacco and nicotine-containing products.

## Figures and Tables

**Figure 1 ijerph-19-05623-f001:**
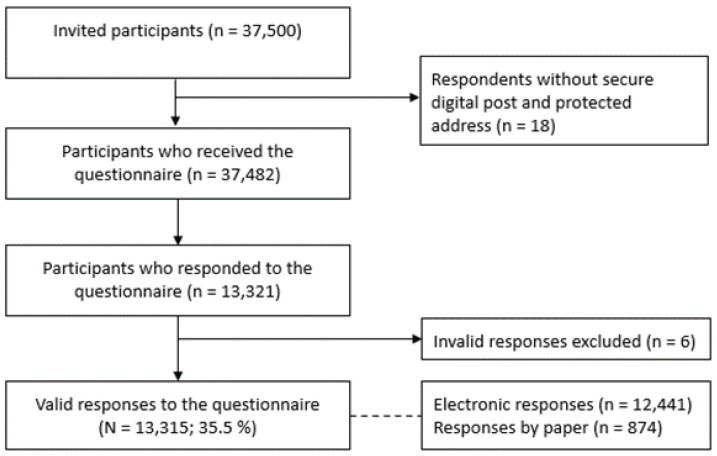
Flow diagram over participants in the §SMOKE study.

**Figure 2 ijerph-19-05623-f002:**
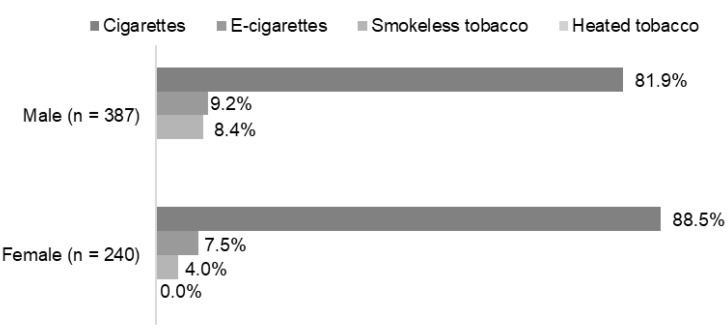
Type of tobacco or nicotine product used first, stratified by gender. Data for the use of heated tobacco among men is not shown due to too few respondents (*n* < 5).

**Figure 3 ijerph-19-05623-f003:**
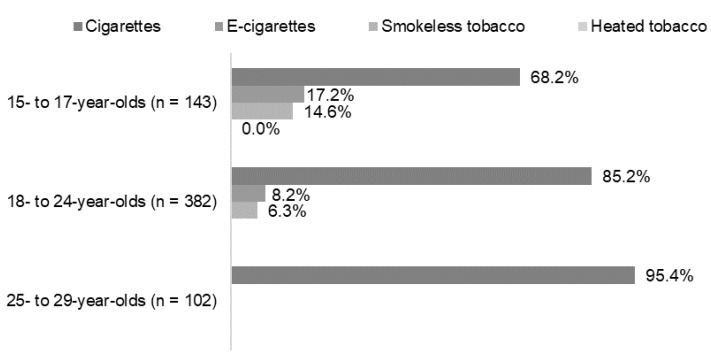
Type of tobacco or nicotine product used first, stratified by age groups. Data for the use of heated tobacco among 18–24-year-olds as well as data for the use of e-cigarettes, smokeless tobacco, and heated tobacco among 25–29-year-olds are not shown due to too few respondents (*n* < 5).

**Table 1 ijerph-19-05623-t001:** Sociodemographic characteristics of the study population.

	Respondents % (*n*)	All Participants Invited to Study % (*n*)
**Gender**		
Men	44.4 (5914)	51.3 (19,222)
Women	55.6 (7401)	48.7 (18,260)
**Age**		
15–17 years	21.8 (2906)	18.0 (6754)
18–24 years	48.2 (6421)	47.0 (17,596)
25–29 years	30.0 (3988)	35.0 (13,132)
**Region of residence**		
Capital Region	32.2 (4288)	34.3 (12,857)
Region Zealand	11.5 (1530)	11.6 (4345)
Region of Southern Denmark	20.6 (2743)	19.7 (7390)
Central Denmark Region	25.5 (3396)	24.2 (9064)
North Denmark Region	10.2 (1358)	10.2 (3826)

**Table 2 ijerph-19-05623-t002:** Measures of current tobacco or nicotine product use, and first product used.

Variable	Item	Response Categories	Coding
**Cigarettes**	Do you smoke cigarettes?	(1) Yes, everyday (2) Yes, at least once a week (3) Yes, less often than every week (4) No, but I have previously smoked/used (5) No, but I have tried smoking/using (6) No, I have never tried smoking/using	Daily (1) Occasionally (2 + 3) Former (4) Tried (5) Never (6)
**E-cigarettes**	Do you use e-cigarettes?		
**Smokeless tobacco**	Do you use any of these…? (a) Snus (b) Chewing tobacco (c) Nicotine bags		
**Heated tobacco**	Do you smoke heated tobacco? (Also known as tobacco sticks, heat sticks etc.—used in a heating device)		
**Product first used**	If you use multiple tobacco products: Which product did you try first?	(1) Cigarettes (2) Smokeless tobacco (3) E-cigarettes (4) Heated tobacco	

**Table 3 ijerph-19-05623-t003:** Use of tobacco or nicotine products by gender and age group, *p*-value for differences by gender and age group.

	Overall % (*n*)	Gender % (*n*)			Age Group % (*n*)			
		Male	Female	*p*	15–17 Years	18–24 Years	25–29 Years	*p*
**Cigarettes**								
Daily	9.8 (1171)	10.4 (546)	9.2 (625)	0.002	4.1 (112)	10.7 (643)	11.5 (416)	<0.001
Occasionally	10.3 (1278)	11.1 (606)	9.6 (672)		7.8 (215)	12.4 (741)	8.9 (322)	
Tried	33.8 (4154)	32.7 (1764)	34.8 (2390)		24.8 (675)	34.5 (2081)	37.4 (1398)	
Former	10.2 (1204)	10.0 (507)	10.5 (696)		2.6 (71)	9.2 (556)	15.5 (577)	
Never	35.9 (4643)	35.9 (2046)	35.9 (2597)		60.7 (1651)	33.2 (1993)	26.8 (999)	
**E-cigarettes**								
Daily	1.8 (212)	2.5 (130)	1.2 (82)	<0.001	1.2 (30)	2.1 (119)	1.8 (63)	<0.001
Occasionally	2.1 (255)	2.5 (141)	1.6 (114)		3.3 (85)	2.2 (128)	1.2 (42)	
Tried	31.6 (3838)	34.0 (1831)	29.1 (2007)		26.5 (701)	37.5 (2186)	26.6 (951)	
Former	6.8 (799)	8.9 (474)	4.8 (325)		4.3 (111)	8.3 (471)	6.2 (217)	
Never	57.7 (7085)	52.2 (2754)	63.3 (4331)		64.7 (1741)	49.9 (2979)	64.1 (2365)	
**Smokeless tobacco**								
Daily	4.3 (495)	6.7 (373)	1.7 (122)	<0.001	3.6 (90)	6.0 (327)	2.4 (78)	<0.001
Occasionally	4.8 (570)	6.5 (355)	3.0 (215)		5.1 (133)	5.9 (335)	3.1 (102)	
Tried	27.3 (3243)	28.4 (1462)	26.3 (1781)		17.5 (469)	28.9 (1701)	30.4 (1073)	
Former	3.6 (426)	5.0 (268)	2.2 (158)		3.4 (88)	4.5 (251)	2.6 (87)	
Never	60.0 (7439)	53.4 (2867)	66.7 (4572)		70.4 (1889)	54.8 (3258)	61.6 (2292)	
**Heated tobacco**								
Daily	0.1 (14)	0.1 (6)	0.1 (8)	<0.001	n/a	0.2 (9)	n/a	<0.001
Occasionally	0.2 (24)	0.3 (16)	0.1 (8)		n/a	0.3 (15)	n/a	
Tried	3.2 (353)	3.8 (192)	2.5 (161)		1.1 (29)	2.8 (159)	4.7 (165)	
Former	0.4 (44)	0.6 (31)	0.2 (13)		0.2 (6)	0.4 (24)	0.4 (14)	
Never	96.1 (11,728)	95.2 (5073)	97.1 (6655)		98.4 (2614)	96.4 (5664)	98.4 (3450)	
**Using at least one product**	27.0 (3219)	31.1 (1657)	22.8 (1562)	<0.001	18.1 (476)	31.3 (1833)	25.8 (910)	<0.001
**Ever used or tried any product**	71.3 (8730)	72.4 (3901)	70.2 (4829)	0.007	50.4 (1370)	74.4 (4467)	78.1 (2893)	<0.001

**Table 4 ijerph-19-05623-t004:** Dual and multiple use of tobacco or nicotine-containing products.

	Overall % (*n*)	Gender % (*n*)		Age Groups % (*n*)		
		Male	Female	15–17 Years	18–24 Years	25–29 Years
**Dual use (Two products)**	87.2 (597)	85.4 (370)	91.2 (227)	78.6 (121)	88.2 (374)	92.1 (102)
**Cigarettes**						
+ E-cigarettes	34.5 (208)	26.7 (95)	50.9 (113)	30.8 (38)	29.4 (115)	52.0 (55)
+ Smokeless tobacco	61.5 (365)	68.1 (255)	47.4 (110)	62.9 (76)	66.5 (244)	45.8 (45)
+ Heated tobacco	1.1 (6)	1.2 (4)	0.9 (2)	0 (0)	1.0 (4)	2.2 (2)
**E-cigarettes**						
+ Smokeless tobacco	2.6 (16)	3.5 (14)	0.8 (2)	5.3 (6)	2.8 (10)	0 (0)
+ Heated tobacco	0.33 (2)	0.5 (2)	0 (0)	0.9 (1)	0.3 (1)	0 (0)
**Smokeless tobacco**						
+ Heated tobacco	0.0 (0)	0.0 (0)	0.0 (0)	0.0 (0)	0.0 (0)	0.0 (0)
**Multiple use (Three or four products)**	12.8 (89)	14.6 (65)	8.8 (24)	21.4 (33)	11.9 (48)	7.9 (8)

## Data Availability

Data not available.

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
