# Peer review of "Use of Tobacco and Nicotine Products among Young People in Denmark—Status in Single and Dual Use"

_ijerph, 2022, doi:10.3390/ijerph19095623_

Round 1
Reviewer 1 Report
Review of manuscript entilted: “Use of tobacco and nicotine products among young people in Denmark – status in single and dual use” authored by Lotus Sofie Bast, Marie Borring Klitgaard, Simone Gad Kjeld, Nanna Schneekloth Jarlstrup, Anne Illemann Christensen
First of all I want to thank you for opportunity to review this interesting manuscript.
In the presented article, authors describe frequency of use of different types of tobacco-related products among young people in Denmark. Introduction is short and provides sufficient information about undertaken problem. Methods are described sufficiently with a very clear diagram showing sample selection. Results are mostly clearly presented, despite Table 3. which needs some polishing. Discussion and conclusions are well-written, based on obtained results.
Overall, the manuscript is well-written and concerns a very serious and worldwide problem in health. I have only one minor remark, which is worth considering.
Major concerns:
None
Minor concerns:
- Line 107 – typo “se” instead of “see”
- Line 118 – bracket is not closed
- Table 3. I do not see information in the table description if p-value is regarding comparison between male and female groups or between frequency of using tobacco products. Moreover, in relation to age-groups part there is also a problem which groups differ between each other, which is not marked here.
Author Response
Thank you for your positive comments. Please find our responses in the attachment.

Reviewer 2 Report
Dear authors
I have no significant qualms about this study, I think the manuscript is well written and the data well presented.
Author Response
Thank you for this.

Reviewer 3 Report
Adolescent and youth tobacco use is undoubted an important public health topic. This study reported tobacco use in youth in Denmark. The strength of this study is the nationwide study sample and detailed categories for tobacco products. The major problem is the writing.
1. The first paragraph in the introduction is not clear. Sentence 31-36 should add the numbers instead of just saying decrease or increase. Sentences 31-34 are particularly difficult to understand.
2. Numbers in line 39-40 is incorrect, please cite more recent data. Line 41 is not clear whether this number is from the U.S., if yes, then the number is wrong too. Please define “current” in these sentences.
3. Again, please replace ref.10 with a more recent reference. When reported % of use, please be very clear about what does this % means, like, current, lifetime, etc.
4. Still in the introduction, authors should provide numbers to support their idea of a shift from cigarette to e-cigarette compensated the decreased use of traditional cigarettes, therefore the overall tobacco use remains at the same level.
5. Add $ after DKK will help readers outside of Denmark to understand the prices.
6. In the study design, line 83 needs modification. Since data of 2020 was collected in Feb-Mar 2020, assume that data of 2021 is already completed at the time of writing this paper.
7. In table one, the response option 4) and 5) are very similar. What do authors do to make sure participants understand these two items correctly? What is the criteria for former use and what is for tried?
8. Also in table one, responders answer the question referring to what time period? For example, past-month?
8. line 135 is incomplete, numbers of five or less of what?
9. Table 2 should add statistic analysis to show whether the response rate is similar between genders, age groups, and regions.
10. Results need to be re-written. Currently, authors just repeat the contents for the table in text.
11. Consider adding sex-by age group interaction for table 2 and table 3
12. In Figures 2 and 3, why do some figures only have 3 categories? The colors scales are too closed.
13. In the discussion, the authors repeated the results again.
14. Consider re-organize the text and figures, making text clear but succinct, and figures more readable.
Author Response
Thank you for good comments. Please find point by point responses in the attachment.

Reviewer 4 Report
Congratulations for a very interesting article. It will help policy-makers to enforce tobacco control policies at country level.
Please provide the 95% Confidence Interval for table 3 and 4 and figure 2.
Author Response
Thank you for the positive comments.

Reviewer 5 Report
Abstract:
Lines 8-9: would you please be more specific? E.g., could you provide some statistics instead of a general statement? Also, would referring to adverse effects of tobacco use justify the urgent need to follow youth tobacco product use better than referring to number of new products?
Line 10: please use past tense.
Line 11: what was meant by dual use?
Lines 12-13: sentence on the number of individuals who received the questionnaire and response rate can be deleted from the abstract.
Introduction:
Lines 24-30: would you please elaborate if the trends were the same across high- vs. low-income countries? Also, if the use of e-cigarettes has been increasing has the use of conventional cigarettes been decreasing?
Line 28: what is meant by “a political” vs. “societal point of view”?
Line 32: prevalence should be singular if referred to the decrease in smoking of conventional cigarettes.
Line 33: it may be more informative to specify in which groups of the Danish youth an increase in cigarette smoking uptake has occurred and over which timeframe, especially given the contents of the next sentence.
Line 38: please, consider replacing the term “Westernized” with other characteristics of countries, e.g., income level, or another opening phrase altogether.
Line 42: What is meant by “a country like Denmark in several ways”? E.g., culturally, economically. It may be more informative for readers who have not had the pleasure of living or visiting Denmark (or Norway) or not otherwise familiar with the countries in general?
Lines 48-49: would you please elaborate on the individual and the overall public health effects from use of any tobacco products and provide reference(s)?
Lines 51-53: please support your statement with a reference.
Lines 57-59: please consider modifying your sentence along the following lines, “Although for years, tobacco regulations have been lenient in Denmark, a new law went into effect.”
Line 64: ban on (instead “of”).
Line 68: in response to (instead of “as a response”).
Line 70: does the project involve a cohort design or measurement overtime?
Line 72: Is this study based on data from the SMOKE study? Would you please clarify in the manuscript? It is not clear until line 79.
Also, it appears that the SMOKE study might have been initiated in response to the recently passed laws and recognition of importance to track/evaluate their effectiveness. If so, describing it may strengthen significance of your study.
Furthermore, description of global surveillance systems on tobacco control and whether they report data on Danish youth or not may strengthen significance of your study as well. For example, ESPAD. You could compare participation/response rates between the surveys (e.g., 2019 ESPAD class/school participation rates for Denmark were 21% -- lower than in your study) and age groups covered by the survey. The GYTS is not conducted in Denmark, unlike in many other countries.
You could also describe progress of Denmark in terms of MPOWER measures. Surveillance of tobacco use, especially novel products, is essential for continued progress. Hence, significance of SMOKE study and resulting prevalence estimates. https://www.who.int/teams/health-promotion/tobacco-control/global-tobacco-report-2021
Line 99: please remove “the last.”
Line 105: can replace “point-of-sale” with POS (used earlier).
Line 117-118: please close the bracket, ie ), after [14].
Lines 127-137: please consider re-organizing this section to refer to descriptive analyses first and then, bivariate analyses.
Line 134: please replace STATA with Stata.
Line 139: the sentence on participation rate can be deleted; this information has been reported earlier in the manuscript. The information on lines 140-142 should be moved to the section on participants and sample size.
Since weighted analyses were conducted using data obtained via probability sampling, the results need to be reported in terms of the study population (rather than study participants). Please ensure correct verbiage throughout the manuscript, especially the Results.
Is the purpose of Table 2 to compare characteristics of respondents with those of all participants invited to the study? If so, this information can be reported in a supplement or briefly summarized in the section on participants and sample size. In the manuscript, it would be most helpful for the reader to understand distribution of characteristics of the study population, and therefore, please consider reporting weighted percentages with corresponding 95% CI or standard errors). The unweighted sample size(s) can be reported in the table footnotes.
Please ensure of sequence of tenses when reporting results.
Lines 158-159: the first part of the sentence refers to gender differences, while the last – to age differences. Please reconcile.
Table 3: similar to my comments for Table 2, please replace unweighted number of observations with 95% CI or standard errors.
Also, please conduct post-hoc comparisons to identify and report specific differences in cigarette, e-cigarette, smokeless tobacco, and heated tobacco use by gender.
Lines 171 – 176: please elaborate in the methods how the number of products used was calculated (e.g., by summation of response options across types of products and frequency use). Also, was the most common combination examined based on that number (e.g., out of 16 possible combinations)?
Table 4: please see my comments for Table 3.
Figures 2 and 3: please add error bars. Heated tobacco can be removed from Figure 2. Also, are both figures necessary?
Line 186: “… age group stated …” (instead of “had stated”).
Discussion
Line 198: please remove “still”.
Lines 243-246: when referring to ever-use, would you please remind the readers if it includes all products in the current and cited studies?
Lines 239-240: you conclude that a significant proportion of young people in Denmark have tried smokeless tobacco relative to other Nordic countries. However, in previous sentences you refer to statistics from 17 other European countries: how many of them were Nordic countries?
Perhaps, it is possible to report an average specific to Nordic countries (assuming it is reported in the cited study by Leon et al.).
Lines 254 – 256: the study design (cohort or panel, whether it is a cohort study or panel – please clarify in the methodology) is a strength (compared to a one-time survey).
Lines 258-259: can be removed. Nationally representative data and study design are important strengths. Was the number of observations sufficient to explore dual and multi-use in more details, e.g., by regions, age, gender? If so, it could be a strength as well, since many common surveillance systems are limited in terms of the sample size.
Line 273: have you come across more recent references or validation studies that you can site?
Lines 274: since your study is descriptive, the cross-sectional design is not a limitation. Instead, I would highlight significance of your study in terms of providing baseline nationally representative statistics (if you can describe in the introduction that such data are not reported by other surveillance systems for tobacco control or other sources, e.g., from sales of tobacco products).
Line 279: “with”
Lines 283 – 285: great idea: without such linkage it will not be possible to study adverse health effects, e.g., cancer and other diseases with lag time, from e-cigarette use in youth.
Author Response
Thank you for your many good suggestions to our manuscript. We hope that you will find merit in the new version.

Round 2
Reviewer 3 Report
Thank you for authors taking time to revise the manuscript. I have the following questions:
- As I mentioned for the reference 10, here is the more recent one: https://www.cdc.gov/mmwr/volumes/71/ss/ss7105a1.htm which shows "In 2021, an estimated 34.0% of high school students (5.22 million) and 11.3% of middle school students (1.34 million) reported ever using a tobacco product... Current (past 30-day) use of a tobacco product was 13.4% (not 31.2% in the manuscript) for high school students.....E-cigarettes ... cited by 11.3% (not 27.5% in the manuscript) of high school students (1.72 million) and 2.8% of middle school students (320,000), followed by cigarettes, cigars ". Authors should justify why used an older reference.
- In the table 4, seems that women were more likely to use both cigarettes and e-cigarette. Since this manuscript would like to focus on non-traditional tobacco products, I think this finding worth to highlight in the result and maybe discussion as well
Author Response
Thank you for providing good comments once again. Please find responses in the attachment.
